# Recycling of Industrial Waste Gypsum Using Mineral Carbonation

**Chan-Ung Kang** [1], **Sang-Woo Ji** [2] **and Hwanju Jo** [1,*]

1 Climate Change Response Division, Korea Institute of Geoscience and Mineral Resources (KIGAM), 124, Gwahak-ro, Yuseong-gu, Daejeon 34132, Korea; cukang1001@kigam.re.kr
2 Policy & Planning Division, Korea Institute of Geoscience and Mineral Resources (KIGAM), 124, Gwahak-ro, Yuseong-gu, Daejeon 34132, Korea; swji@kigam.re.kr
* Correspondence: chohwanju@kigam.re.kr

**Abstract:** Direct mineral carbonation (MC) is used to mitigate carbon dioxide ($CO_2$) emissions. This method has the great advantages of reducing the amount of industrial residues and creating valuable materials by incorporating $CO_2$. Waste gypsum, industrial waste including flue gas desulfurization (FGD) gypsum (25.27–53.40 wt% of CaO), and phosphogypsum (30.50–39.06 wt% of CaO) can be used for direct MC (conversion rate up to 96%). Mineral carbonation converts waste gypsum into calcium carbonate ($CaCO_3$), which can be recycled during desulfurization. Furthermore, ammonium sulfate (($NH_4$)$_2SO_4$), which is used as a fertilizer, can be prepared as a by-product when the carbonation reaction is performed using ammonia ($NH_3$) as a base. In this study, recent progress in the carbonation kinetics and preparation of $CaCO_3$ using FGD gypsum and phosphogypsum with $NH_3$ was investigated. Temperature, $CO_2$ partial pressure, $CO_2$ flow rate, and $NH_3$ concentration were reviewed as factors affecting carbonation kinetics and efficiency. The factors influencing the polymorphs of the prepared $CaCO_3$ were also reviewed and summarized. A state-of-the-art bench-scale plant study was also proposed. In addition, economic feasibility was investigated based on a bench-scale study to analyze the future applicability of this technology.

**Keywords:** mineral carbonation; waste gypsum; calcium carbonate; ammonia

## 1. Introduction

Mineral carbonation (MC) plays an important role in carbon dioxide ($CO_2$) storage. Most MC technological developments are still focused on ex situ process concepts. Since the 1990s, scientists have begun fundamental research on MC using peridotite to accelerate natural processes [1]. Recently, several studies using byproducts have been reported. The ability of MC with industrial waste is limited and much less than that of MC with silicate minerals [2,3]. However, studies on MC with industrial waste have reported positive results. Most industrial wastes can acquire acceptable MC efficiencies under mild conditions [4]. Industrial wastes containing divalent metals generally have a higher chemical reactivity, in addition to the requirement that the problem of industrial waste be treated as waste [4]. Therefore, MC from industrial waste addresses waste problems and achieves acceptable $CO_2$ conversion under mild conditions [3,5]. Industrial waste can come from various sources such as coal fly ash [6–8], metallurgical slag [9,10], waste concrete [11,12], mine waste [13,14], and industrial waste gypsum [15,16].

Waste gypsum, among the industrial wastes, has a high calcium (Ca) content; therefore, it has good evaluation potential for $CO_2$ capture. In the case of flue gas desulfurization (FGD) gypsum, the flue gas of the power plant can be used directly, and desulfurized gypsum, the starting material, can be continuously supplied. This eliminates the cost of $CO_2$ capture, transportation, and raw materials. In addition, both calcium carbonate ($CaCO_3$) and ammonium sulfate (($NH_4$)$_2SO_4$) generated after the reaction can be commercialized for use as $CaCO_3$ for desulfurization and fertilizers (Figure 1). Phosphogypsum, generated

by reacting sulfuric acid and phosphate rock to produce phosphoric acid [17], also has potential for MC. The annual production of FGD gypsum and phosphogypsum exceeds 140 million tons [18,19] and 280 million tons [20], respectively. If the total amounts of FGD gypsum and phosphogypsum were used to sequester $CO_2$ through MC, almost 100 million tons of $CO_2$ would be stored annually.

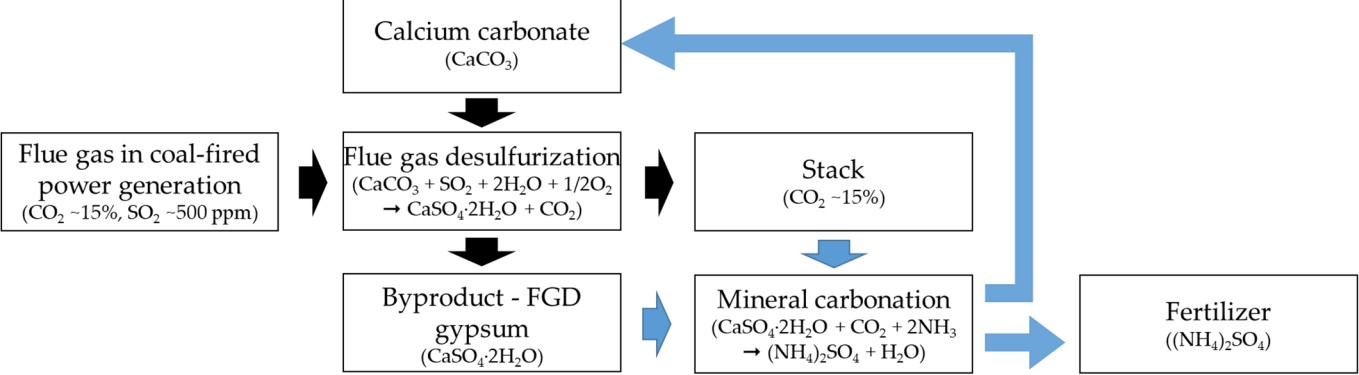

**Figure 1.** Circulation of desulfurized gypsum in coal-fired power plants through application of mineral carbonation (MC) technology (Gas composition was referenced in Lee et al. [21], KIGAM [22]).

Waste FGD gypsum and phosphogypsum usually take the form of calcium sulfate dihydrate ($CaSO_4 \cdot 2H_2O$). Thus, these gypsums can be used to produce $CaCO_3$ through direct carbonation by injecting $CO_2$ gas into a suspension of waste gypsum and ammonia ($NH_3$) solution [23,24]. Ammonium hydroxide ($NH_4OH$) is used to enhance the MC effect. As $NH_4OH$ is a readily available basic source for many industrial processes, the utilization of $NH_4OH$ is attractive from an environmental point of view [25].

The prepared $CaCO_3$ using byproduct gypsum as a raw material not only realizes the resource utilization of byproduct gypsum, but also saves the natural resources of $CaCO_3$. Furthermore, in the direct carbonation of by-product gypsum, a highly pure sulfate salt can be obtained through crystallization using the difference in solubility with $CaCO_3$ (Equation (1)) [26,27]. A stoichiometric excess $NH_3$ and $CO_2$ react to form an ammonium compound. The decomposition temperature of these ammonium compounds is 60 °C or less, which can be easily recovered during the crystallization process of (($NH_4)_2SO_4$) aqueous solutions.

$$CaSO_4 \cdot 2H_2O(s) + CO_2(g) + 2NH_4OH(aq) \rightarrow (NH_4)_2SO_4(aq) + CaCO_3(s) + 2H_2O \quad (1)$$

Several researchers have recently reported developments in this field. Parameters affecting direct aqueous MC and indirect aqueous MC using industrial byproduct gypsum, including FGD gypsum, phosphogypsum, and red gypsum, have been analyzed by Wang et al. [25]. An analysis of the scale-up applications of MC using industrial solid waste was also reported by Liu et al. [28].

The technology of MC using waste gypsum has been advanced through several lab scale studies. However, there have been limited discussions on bench-scale studies of MC by reacting flue gas and $NH_3$ with waste gypsum, and a detailed economic analysis has not been reported. This makes comparison with other MC technologies difficult and does not guarantee field applicability of this technology.

This study focused on direct aqueous carbonation using waste gypsum with $NH_3$. The state-of-the-art bench-scale plant studies were discussed to confirm the potential for scale-up of the technology; in particular, economic feasibility based on the bench-scale studies was investigated to analyze the future applicability of this technology. Furthermore, recent progress in the carbonation performance of two types of byproduct gypsum, FGD gypsum and phosphogypsum, was discussed based on $CO_2$ carbonation efficiency. The factors influencing the carbonation kinetics and preparation of $CaCO_3$ were reviewed.

## 2. Characterization of Waste Gypsum

The compositions of FGD gypsum and phosphogypsum were analyzed. From the results shown in Tables 1 and 2, waste gypsum from different sources differs with respect to its constituents. The main components of the FGD gypsum are sulfur trioxide ($SO_3$), calcium oxide (CaO), and bound water, and the main impurities are silicon dioxide ($SiO_2$), aluminum oxide ($Al_2O_3$), and ferric oxide ($Fe_2O_3$). The composition of phosphogypsum production depends on the quality of the phosphate rock used to produce phosphoric acid ($H_3PO_4$) and the processing route [29].

The major constituents of phosphogypsum are similar to those of FGD gypsum, but contain more impurities such as $SiO_2$, free $H_3PO_4$, phosphates, and organic matter [30]. The sulfur content of FGD gypsum and phosphogypsum was 12.54–22.39 wt% and 12.40–22.80 wt%, respectively, showing similar ranges. However, the different Ca content ranges indicated the difference between these two waste gypsums. FGD gypsum has a wide range (standard deviation 5.55 wt%) with Ca 18.06–38.16 wt%, whereas phosphogypsum has a narrow range (standard deviation 2.15 wt%) with 21.80–27.92 wt% Ca. FGD gypsum is generated during the desulfurization process of flue gas and, as $CaCO_3$ added in a large amount stoichiometrically remains unreacted, the Ca content is higher than the S content. The moisture contents of FGD gypsum and phosphogypsum are usually in the range of 1–23% and 8–30% [31], respectively.

**Table 1.** Oxide composition of flue gas desulfurization (FGD) gypsum for different studies.

| Reference | [32] | [33] | [34] | [26] | [35] | [36] | [37] | [38] | [39] | [40] | [41] |
|---|---|---|---|---|---|---|---|---|---|---|---|
| $SO_3$ | 42.58 | 43.57 | 47.80 | 46.51 | 36.90 | 31.31 | 49.74 | 55.90 | 40.34 | 40.80 | 44.84 |
| CaO | 32.25 | 48.06 | 39.50 | 32.50 | 31.90 | 53.40 | 37.95 | 40.10 | 25.27 | 28.10 | 32.49 |
| $SiO_2$ | 0.78 | 2.95 | 1.05 | - | 3.30 | 2.95 | 2.39 | 2.12 | 1.69 | 2.00 | 0.70 |
| MgO | 0.52 | 1.48 | 0.17 | - | 3.80 | 0.89 | 0.25 | 0.18 | 0.98 | 1.00 | - |
| $Fe_2O_3$ | 0.32 | 0.58 | 0.30 | 0.43 | 0.30 | 0.54 | 0.26 | 0.29 | 0.18 | 0.50 | 0.22 |
| $Al_2O_3$ | 0.12 | 1.18 | 0.93 | 0.56 | 1.00 | 0.91 | 1.43 | 1.23 | 0.34 | 1.20 | 0.42 |
| SrO | - | - | - | 0.04 | - | - | - | - | - | - | - |
| $K_2O$ | - | - | 0.09 | - | - | - | - | - | 0.03 | 0.10 | 0.10 |
| $Na_2O$ | - | - | - | - | - | 0.59 | - | - | - | 0.30 | 0.13 |
| $TiO_2$ | - | 0.39 | - | 0.02 | 0.05 | - | - | - | 0.02 | 0.07 | - |
| CuO | - | - | - | 0.02 | - | - | - | - | - | - | - |
| MnO | - | 1.18 | - | 0.03 | - | - | - | - | - | 0.01 | - |
| $P_2O_5$ | - | 0.03 | - | - | 0.01 | - | - | - | - | - | 0.08 |
| Cl | - | - | 0.077 | - | - | - | 0.26 | - | - | - | - |
| Other | - | 0.58 | - | - | - | - | - | - | - | - | - |
| LOI | 18.34 | - | 10.08 | 19.70 | 22.40 | 8.41 | 7.72 | - | - | - | 20.18 |
| * $CaSO_4$ | 72.40 | 74.09 | 81.28 | 79.09 | 62.75 | 53.24 | 84.58 | 95.05 | 68.59 | 69.38 | 76.25 |
| ** $CaCO_3$ | 4.33 | 31.31 | 10.74 | - | 10.81 | 56.17 | 5.55 | 1.69 | - | - | 1.93 |

* $CaSO_4$ content is calculated assuming that all sulfur constitutes gypsum. ** $CaCO_3$ content is calculated from the Ca content remaining after composing $CaSO_4$.

**Table 2.** Oxide composition of phosphogypsum for different studies.

| Reference | [42] | [43] | [44] | [45] | [46] | [47] | [48] | [49] | [50] | [51] | [30] | [52] | [53] | [54] | [55] | [56] |
|---|---|---|---|---|---|---|---|---|---|---|---|---|---|---|---|---|
| Constituent (wt%) | | | | | | | | | | | | | | | | |
| $SO_3$ | 56.68 | 52.71 | 53.48 | 52.56 | 30.95 | 42.83 | 34.51 | 43.80 | 51.53 | 44.47 | 42.90 | 42.10 | 46.02 | 56.92 | 49.89 | 44.40 |
| CaO | 38.39 | 39.06 | 38.60 | 32.83 | 31.05 | 31.00 | 32.14 | 30.70 | 36.60 | 30.52 | 30.50 | 31.64 | 32.12 | 33.64 | 36.48 | 32.80 |
| $SiO_2$ | 1.37 | 0.34 | 0.37 | 12.87 | 4.86 | 0.13 | 8.82 | 1.38 | 0.61 | 5.05 | 9.50 | 4.86 | 5.94 | 6.31 | 1.89 | 1.37 |
| MgO | 0.12 | 0.21 | 0.04 | 0.08 | 0.26 | - | 0.09 | 0.01 | - | - | 0.30 | 0.29 | 0.30 | - | - | 0.01 |
| $Fe_2O_3$ | 0.45 | 0.04 | 0.14 | 0.10 | - | 0.50 | 0.35 | 0.02 | - | 0.35 | 0.90 | 0.27 | 1.54 | 0.33 | - | 0.03 |
| $Al_2O_3$ | 0.35 | 0.07 | 0.13 | 0.55 | - | 0.77 | 0.29 | 0.10 | 0.18 | 0.66 | 2.80 | 4.38 | 0.50 | 0.64 | 0.08 | 0.11 |
| $K_2O$ | 0.12 | - | - | 0.16 | 0.41 | - | - | - | 0.02 | 0.14 | - | - | - | - | - | - |
| $Na_2O$ | 0.07 | - | - | 0.03 | - | - | - | 0.06 | 0.13 | 0.08 | - | 0.17 | - | - | - | - |
| $TiO_2$ | 0.06 | - | - | 0.07 | 0.20 | - | - | - | - | 0.07 | - | - | - | - | - | - |
| $P_2O_5$ | 2.26 | 1.11 | 0.82 | - | 3.57 | 3.91 | 1.72 | 2.51 | 0.33 | 0.81 | 0.50 | 1.05 | 1.39 | 0.71 | 0.44 | 1.69 |
| $CO_2$ | - | - | - | - | - | - | - | - | - | - | 4.50 | - | - | - | - | - |
| F | - | 0.06 | 0.14 | - | - | 0.26 | 0.80 | 1.93 | - | 0.26 | 0.15 | 0.12 | - | 0.91 | - | - |
| LOI | - | 6.40 | 6.40 | 11.20 | 22.91 | 20.00 | 21.00 | - | - | 18.29 | 3.50 | 21.19 | - | - | 11.22 | 22.30 |
| * $CaSO_4$ | 96.38 | 89.63 | 90.94 | 89.37 | 52.63 | 72.83 | 58.68 | 74.48 | 87.62 | 75.62 | 72.95 | 71.59 | 78.25 | 96.79 | 84.83 | 75.50 |
| ** $Ca_3(PO_4)_2$ | - | 2.43 | 1.79 | - | 7.80 | 8.54 | 3.76 | 5.48 | 0.72 | - | 1.09 | 2.29 | - | - | 0.96 | 3.69 |

* $CaSO_4$ content is calculated assuming that all sulfur constitutes gypsum. ** $Ca_3(PO_4)_2$ content is calculated from the Ca content remaining after composing $CaSO_4$.

## 3. Mineral Carbonation of Byproduct Gypsum

### 3.1. Carbonation Kinetics and Preparation of CaCO$_3$

Several factors influence the carbonation kinetics and preparation of CaCO$_3$. Herein, these factors, including temperature, NH$_3$ concentration, solid–liquid ratio, CO$_2$ flow rate, CO$_2$ partial pressure, induction period, kinetics, and preparation of CaCO$_3$ are discussed.

#### 3.1.1. Temperature

Gypsum carbonation is a highly exothermic reaction that rapidly increases the temperature of the reaction slurry in a short time. As gypsum carbonization is an exothermic process, high temperatures are undesirable for carrying out carbonization from a thermodynamic point of view. However, the temperature can affect the rate of carbonation. The conversion of gypsum to CaCO$_3$ increases with elevated temperatures. A study showed that carbonation of FGD gypsum at 40 °C was nearly twice as fast as that at room temperature [41]. FGD gypsum conversion was close to 90% at 40 °C after 1 h, according to the results of another study, and the conversion reached up to 100% at a high temperature of 80 °C (Figure 2) [57].

However, studies have shown that the efficiency decreased gradually with increasing temperature due to the decrease in the solubility of CO$_2$ and the decomposition of ammonium salts [41,58]. At ambient pressure, the carbonation conversion efficiency decreased with increasing temperature. The conversion efficiency of FDG gypsum carbonation at 40 °C (96%) was almost the same as at room temperature [41]. At temperatures above 60 °C, the efficiency was significantly lowered, and the carbonation conversion efficiency at 80 °C was approximately 20% lower than that at 40 °C [41]. This is the result of FGD gypsum not completely converted to CaCO$_3$ owing to NH$_4$OH loss at high temperatures of 60 °C and 80 °C [58]. However, below 40 °C, the temperature did not significantly affect the efficiency [41]. The conversion efficiency curves of phosphogypsum at 30 °C and 80 °C were shown to be similar with 4 bar and pure CO$_2$ gas. However, the carbonation conversion efficiency of phosphogypsum increased slightly as the temperature increased from 30 °C to 80 °C when the pressure was increased to 8 bar [54].

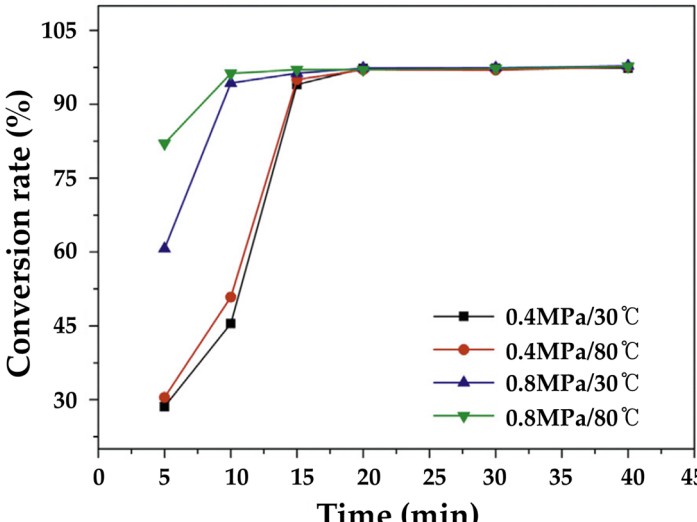

**Figure 2.** Conversion rate as a function of initial temperature (Reprinted/adapted with permission from [54]. 2015, Elsevier).

Compared with phosphogypsum, FGD gypsum showed better reaction activity under normal conditions. It took approximately 30 min to achieve 90% conversion of FGD gypsum at 40 °C and less than 1 h for phosphorus gypsum at room temperature and atmospheric pressure [59].

The time required to reach reaction equilibrium is longer at low temperatures than at high temperatures [57]. Therefore, an appropriate temperature must be selected to balance the thermodynamics and kinetics of carbonation to achieve optimal $CO_2$ sequestration efficiency.

### 3.1.2. $CO_2$ Partial Pressure

Studies have found that efficiency may vary depending on the pressure because $CO_2$ dissolution is the rate-determining step in direct carbonation [26,27]. At $CO_2$ pressures higher than atmospheric conditions, more carbonate ions ($CO_3^{2-}$) were produced, promoting carbonation. Furthermore, the high partial pressure of $CO_2$ inhibits the decomposition of ammonium (bi)carbonate ($NH_4HCO_3$) when $NH_4OH$ is used as the basic medium [25].

Furthermore, despite the different initial temperatures (30 °C, 80 °C), the carbonation conversion ratio was similar at 0.4 MPa, but when the pressure was increased to 0.8 MPa, a temperature difference appeared. Accordingly, the phosphogypsum carbonation reaction rate is more sensitive to temperature at high pressure (0.8 MPa) than at low pressure (0.4 MPa) [54].

The carbonation conversion ratio increased significantly as the pressure increased from 0.4 MPa to 0.8 MPa (Figure 2). In addition, because a high carbonation rate reduces $NH_4OH$ loss, $CO_2$ partial pressure has a significant effect on carbonation [54]. It was found that FGD gypsum achieved more than 90% carbonation conversion in 20 min at 60 mL/min at atmospheric pressure, whereas a comparable carbonation conversion (95%) for phosphogypsum required an increase in the $CO_2$ pressure [54].

As $CO_2$ dissolution and the dissociation of carbonic acid are rate-limiting steps in the aqueous carbonation process, the rate of the carbonation reaction can be strengthened by higher pressures [27,60]. However, increasing the pressure increases the cost, and an appropriate partial pressure must be used to enhance the MC.

### 3.1.3. $CO_2$ Flow Rate

Theoretically, the total $CO_2$ uptake should remain the same. However, at various $CO_2$ flow rates, the calculated saturated $CO_2$ uptakes at the inflection points of the curves were shown to be different. At a $CO_2$ flow rate of 60 mL/min, the inflection point occurred at 20 min, when saturated $CO_2$ sequestration had already been achieved. The total $CO_2$ uptake, that is, the uptake capacity, can be attributed to the flow rate, inflection point time, and conversion rate. Therefore, carbonation efficiency increases with increasing flow rate and $CO_2$ content [57]. Equilibrium was reached three times faster when the $CO_2$ flow rate was 4 L/min than 1 L/min (Figure 3) [21]. When using 100 vol% $CO_2$, more than 94% of the phosphogypsum was converted to calcium carbonate in approximately 30 min, whereas using 20 vol% $CO_2$ required 90 min to achieve similar levels [61]. However, the $CO_2$ sequestration efficiency did not always increase with the $CO_2$ flow rate. The conversion increased slightly, but the time to reach reaction equilibrium became significantly shorter as the $CO_2$ flow rate increased [62]. The optimal $CO_2$ flow rate should therefore be selected based on $CO_2$ storage efficiency, carbonation time, and product quality.

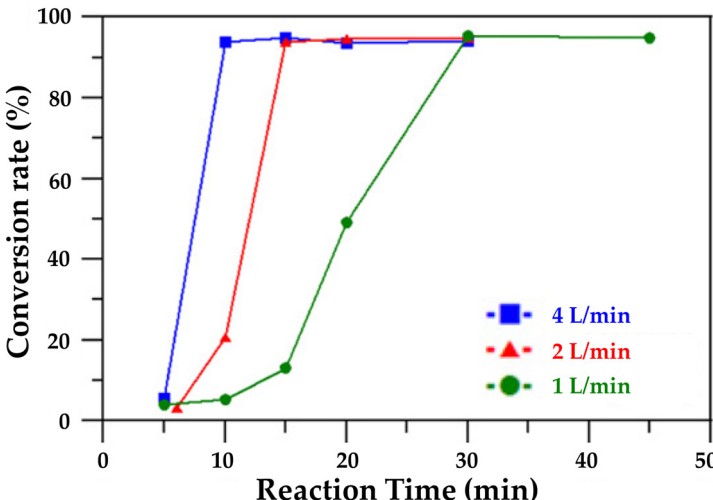

**Figure 3.** Conversion rate as a function of $CO_2$ flow rate (Reprinted/adapted with permission from [21]. 2012, Elsevier).

### 3.1.4. NH$_3$ Concentration

$NH_3$ can increase the carbonation rate and produce $((NH_4)_2SO_4)$ as a byproduct of the carbonation process. The conversion efficiencies with $NH_3$ have been reported to be higher than 95% at 10 min [21,54]. The carbonation efficiency of FGD gypsum was shown to increase with increasing $NH_4OH$ concentration, and the carbonation reaction was almost completed within 10 min [21]. Although stoichiometrically the molar ratio of $OH^-/Ca^{2+}$ is 2, $NH_4OH$ must be replenished because $NH_4OH$ can be lost during carbonation at temperatures above 60 °C. High conversion efficiency can be obtained by reacting with a molar ratio of $OH^-/Ca^{2+}$ of 2.1–3.6, higher than the stoichiometric ratio [26,57]. However, the excess ammonia ratio does not control the reaction rate (Figure 4) [54]. The study showed that at a very low $NH_3$ content, the reaction equilibrium was reached within 10 min, but the FGD gypsum conversion was only 43% [57]. The FGD gypsum conversion rate rapidly improved with increasing $NH_3$ content. In addition, it has been reported that an increase in the $CO_2$ partial pressure and $NH_3$ content is more favorable for carbonate ion formation through thermodynamic modeling of an electrolyte non-random two-liquid (NRTL) model of the $NH_3$-$CO_2$-$H_2O$ system [63].

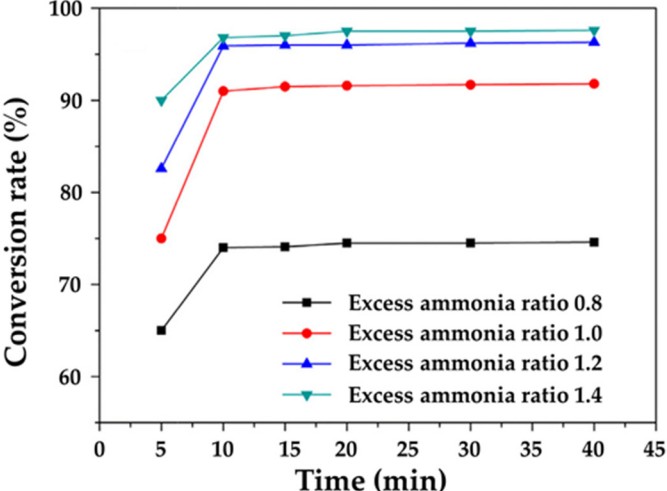

**Figure 4.** Conversion rate as a function of ammonia-to-gypsum ratio (Reprinted/adapted with permission from [54]. 2015, Elsevier).

### 3.2. Properties of CaCO$_3$

Depending on the production conditions, CaCO$_3$ prepared in the direct carbonation process of waste gypsum can form polymorphs of calcite, aragonite, and vaterite. A mixture of calcite and vaterite was observed during the production of the FGD gypsum-NH$_4$OH-CO$_2$ system [21]. Various factors, such as temperature, pH, and impurities, were shown to affect the properties of CaCO$_3$ [64,65].

The carbonation temperature can significantly change CaCO$_3$ polymorphs during gypsum carbonation. A study showed that calcite and vaterite peaks predominate at 20 °C and 40 °C, respectively. At 60 °C, the calcite peak was stronger than the vaterite peak. At 80 °C, an aragonite peak was observed without the vaterite peak [41].

A higher Ca$^{2+}$/CO$_3{}^{2-}$-ratio is a key factor in the formation of calcite [66,67]. Furthermore, it was shown that a rhombohedral form of calcite is formed when only a stoichiometric amount of NH$_4$OH was used, whereas spherical vaterite is formed when NH$_4$OH was used in excess [68]. However, the yield of pure rhombohedral calcite was relatively low (5%) under a short induction period [27]. A longer reaction time (>1 h) at 2 °C and an excess of CO$_2$ (4.2 mol/mol Ca) favored the transformation of vaterite to rhombohedral calcite [69]. It was shown that injected CO$_2$ also affects the formation of polymorph CaCO$_3$ to the same extent as Ca$^{2+}$ and NH$_4$OH. For the phosphogypsum–NH$_4$OH–CO$_2$ system, direct carbonation with NH$_4$OH under pure CO$_2$ resulted in only calcite [54,70]. A mixture of calcite and vaterite was obtained in 20% CO$_2$ [61].

The dolomite in gypsum is an important factor in the generation of CaCO$_3$ polymorphs. The hydrophilicity and negative surface charge of dolomite particles has been shown to play an important role in the selective formation of calcite during FGD gypsum carbonation [34]. The carbonation product of FGD gypsum containing dolomite is a mixture of vaterite (60%) and calcite (40%), and only pure vaterite was obtained in the CaSO$_4$·2H$_2$O carbonation reaction that was performed [34].

Polymorphs of CaCO$_3$ depend on the conditions of the carbonation system. Among the three polymorphs of CaCO$_3$, calcite has the highest stability, whereas aragonite and vaterite are thermodynamically metastable and unstable, respectively [71]. To produce suitable polymorphs according to the utilization plan of the prepared CaCO$_3$, further research on the difference in CaCO$_3$ polymorphs during the carbonation of FGD gypsum and phosphogypsum is required.

### 4. Plant Study

As introduced in the previous section, many studies have proposed and tested strategies on a laboratory scale to overcome the main challenges of MC of waste gypsum. Scale-up studies have been performed on MC using other MC methods or waste materials. However, few efforts have been made to apply direct aqueous MC with NH$_3$ using waste gypsum for scale-up applications.

A representative bench-scale plant study with continuous process was conducted from 2010 to 2015 by the Korea Institute of Geoscience and Mineral Resources (KIGAM) (Figure 5) [22]. Experiments were conducted by manufacturing a bench-scale continuous gypsum carbonation facility, and the reaction system consisted of a bubble column reactor (left of Figure 2) and CO$_2$ and nitrogen (N$_2$) injection facilities (right of Figure 2). The bubble column reactor increased the dissolution efficiency of CO$_2$ such that the flue gas injected into the lower part of the slurry passed through the slurry and moved to the upper part. This study was conducted using FGD gypsum, which has a composition of 32.49 wt% CaO and 44.84 wt% SO$_3$, generated in a thermal power plant, and was used as a raw material for carbonation without pretreatment to improve carbonation reactivity. A slurry with a solid–liquid ratio of 15 wt% containing 120 wt% NH$_3$ based on a chemical equivalent was injected at a rate of 240 kg/h, and 15 vol% of CO$_2$ was injected at a rate of 80 kg/h. Optimization was performed by operating a 1000 ton/year scale facility for 2000 h, resulting in a 97% carbonation rate and crystallization of ((NH$_4$)$_2$SO$_4$).

KIGAM has been conducting empirical research at a scale of 2000 ton/year in with coal-fired power plants since 2021. To reduce $CO_2$ and waste, further field studies beyond laboratory-based experimental and small-scale plant studies should be conducted. This will lead to precise process improvement.

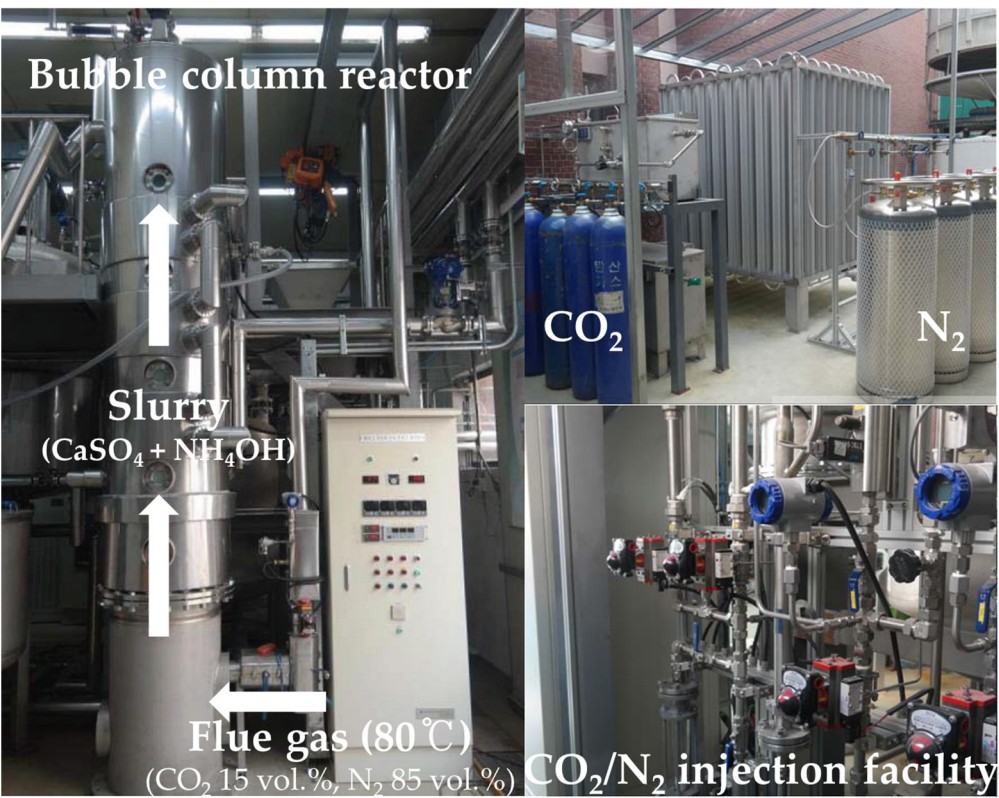

**Figure 5.** Bench-scale continuous gypsum carbonation facility manufactured by KIGAM (**left**—bubble column reactor; **right**—$CO_2$ and nitrogen ($N_2$) injection facilities) [22].

## 5. Techno-Economic Analysis

Although MC has evolved through many previous studies, technology is not the only barrier against deployment; cost also acts as a barrier. Cost penalties are related to plant scale, operating conditions, and operating modulus, such as pre-treatment and/or post-treatment processes [72,73]. For direct carbonation, the energy requirement of the grinding process is the major cost of the overall process [74,75]. The power requirement is 352 MW, nearly 75% of which is the power required for ore grinding operations [74].

The economic feasibility of the carbonation method was analyzed based on the life cycle cost (LCC). LCC is an approach that assesses the total cost of an asset over its life cycle, including initial capital costs, maintenance costs, operating costs, and the asset's residual value at the end of its life. Initial investment cost was calculated through P&ID preparation. Owing to the lack of commercialized plant studies, the cost estimations of accelerated carbonation were based on pilot-scale operations. The simulated MC processes are shown in Figure 1.

This was calculated based on the process of removing $CO_2$ by inputting 300,000 tons of $CaSO_4$ per year. The annual cost was evaluated by setting the lifetime of the initial investment facility to 10–25 years and the real discount rate to 3.2% (Supplementary Information). The results of the calculations are presented in Table 3. Details of the investment cost, operating cost, and operating income can be found in the Supplementary Information.

Operating costs accounted for most of the cost, and $NH_3$ gas, an input raw material, accounted for 58.0% of the cost. The economic feasibility depends on the raw material required for the MC process; therefore, it must be re-evaluated according to the raw material

price. $(NH_4)_2SO_4$, a by-product, may also have different selling prices, which can improve economic feasibility, and should also reflect the sale of carbon credits.

The energy to be used for raw material crushing and the $CO_2$ capture process can be saved, as this technology directly reacts with raw materials without pretreatment with flue gas. This is promising because it is carried out by a single process in an aqueous solution [76,77]. The facility is not structurally complicated; therefore, it is advantageous for scaling up. Depending on the connection with the target plant to be applied, there is the possibility of reducing the steam cost with the operating costs. After the carbonation reaction, a useful material is produced through the crystallization of $(NH_4)_2SO_4$, thereby improving the economic feasibility of MC technology. Although the feasibility study was based on FGD gypsum, it is possible to estimate the carbonation cost for another waste gypsum with gypsum-specific cost adjustments.

**Table 3.** Life cycle costing (LCC) for mineral carbonation (MC) by inputting 300,000 tons of $CaSO_4$ with $NH_3$.

| Initial Investment Cost | Total Converted to Annual Expenses (3.2% of Real Discount Rate) | USD 33,672,343 USD 2,455,073/Year |
|---|---|---|
| Equipment | Devices, heat exchangers, and vessels | USD 19,721,208 |
| | Pump and blower | USD 2,401,836 |
| | Instrument | USD 1,578,514 |
| Construction | Control automation construction, Control panel and electrical construction | USD 861,436 |
| | Plumbing construction | USD 4,090,150 |
| | Structure construction | USD 1,577,629 |
| | Transport and installation work | USD 208,681 |
| | Thermal insulation construction | USD 801,336 |
| | Engineering | USD 2,431,553 |
| **Operating cost** | | **USD 58,703,171/year** |
| Material | Ammonia gas ($NH_3$) | USD 35,477,369/year |
| | Flue gas | USD 0/year |
| | Water | USD 6193/year |
| Utility | Steam | USD 17,981,803/year |
| | Electricity | USD 4,496,312/year |
| | Cooling water | USD 434,798/year |
| Labor costs | | USD 306,696/year |
| **Operating income** | | **USD 42,395,139/year** |
| | Ammonium sulfate ($(NH_4)_2SO_4$) | USD 42,395,139/year |
| **Total** | | **USD 18,736,105/year** |

## 6. Discussion

FGD gypsum and phosphogypsum were discussed to be utilized for MC. These two materials contain 53.24–95.05 wt% and 52.63–96.79 wt% of $CaSO_4$, respectively. The amount of carbonation varies depending on the $CaSO_4$ content. Temperature, $CO_2$ partial pressure, $CO_2$ flow rate, and $NH_3$ concentration have influenced the kinetics and preparation of $CaCO_3$. The kinetics become faster with elevated temperatures. The reaction rate is twice as fast at 40 °C than at room temperature. However, a moderate temperature is recommended because a high temperature accelerates the kinetic while decreasing the amount of $CO_2$ that can be fixed. This reduced efficiency can be compensated for by increasing the $CO_2$ partial pressure. Under high $CO_2$ partial pressure, more $CO_3^{2-}$ is generated, which not only promotes carbonation, but also increases the effect of temperature. It is appropriate to increase the pressure with temperature for an efficient process. $CO_2$ flow rate and $NH_3$ concentration can act as factors that increase MC efficiency. Polymorphs of $CaCO_3$ depend

on the pH, impurities, and $Ca^{2+}/CO_3^{2-}$ ratio along with the above factors, so it is necessary to create appropriate conditions according to the utilization plan of the prepared $CaCO_3$.

Bench scale and pilot scale studies were also conducted using conditions derived from laboratory-scale batch studies. The applicability of the MC technology was improved through the research conducted to directly utilize the flue gas without separating/recovering $CO_2$. When applied to $CO_2$ generating facilities, there is no need for a recovery/separation device because flue gas is directly used. Research conducted as a continuous process can act as a bridgehead for the MC technology to be put to practical use. Moreover, it was shown that direct aqueous MC of waste gypsum using $NH_3$ can also overcome techno-economic hassles. Although the input of $NH_3$ accounts for a significant portion of the operating cost, the sale of ammonium sulfate as a by-product can improve economics.

## 7. Conclusions

Direct aqueous mineral carbonation (MC) of waste gypsum using $NH_3$ is a technology that can help mitigate the global anthropogenic $CO_2$ emissions problem. Recent advances of the technology by many researchers have shown a great potential for $CO_2$ sequestration. The optimal conditions for $CO_2$ sequestration were derived by analyzing the factors affecting the kinetics and preparation of $CaCO_3$: temperature, $CO_2$ partial pressure, $CO_2$ flow rate, and $NH_3$ concentration.

The recently conducted bench scale plant study also proved the field applicability of this technology. Moreover, the cost of sequestering 1 ton of $CO_2$ based on a process that treats 300,000 tons of $CO_2$ is USD 62.5, which is economically analyzed. The economic feasibility of applying this technology to the flue gas generation site has been established. However, more verification through various conditions and process design is required. There is a possibility that the economic efficiency will be further improved according to the $CO_2$ reduction support policy by country and industry.

Before other large-scale $CO_2$ sequestration technologies are put to practical use, it is an important technology that can serve as a bridgehead by utilizing these techno-economic advantages. To this end, a larger-scale empirical study is needed in the future.

**Supplementary Materials:** The following supporting information can be downloaded at: https://www.mdpi.com/article/10.3390/su14084436/s1.

**Author Contributions:** Conceptualization, S.-W.J. and H.J.; investigation, C.-U.K.; data curation, C.-U.K.; writing—original draft preparation, C.-U.K.; writing—review and editing, S.-W.J. and H.J.; supervision, H.J.; funding acquisition, H.J. All authors have read and agreed to the published version of the manuscript.

**Funding:** This work was supported by the Korea Institute of Energy Technology Evaluation and Planning (KETEP) grant funded by the Korea government (MOTIE) (20214710100030, Demonstration of mineral carbonation using FGD gypsum and development of CDM methodology).

**Institutional Review Board Statement:** Not applicable.

**Informed Consent Statement:** Not applicable.

**Data Availability Statement:** Not applicable.

**Conflicts of Interest:** The authors declare no conflict of interest.

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
