# Peer review of "Recycling of Industrial Waste Gypsum Using Mineral Carbonation"

_sustainability, doi:10.3390/su14084436_

Round 1

Reviewer 1 Report

The authors didn't answer the questions raised in the first review round.

CO2 mineral sequestration is one of the promising strategies to abate global warming. This study reviewed recent developments of ex-situ mineral carbonation processes by waste gypsum. It is helpful to understand the key limitation for CO2 mineralization. Although the review dealt with a very important and timely topic, the originality of the review was questionable as its structured very similarly to already existing reviews in the literature. The manuscript needs improvement before it can be considered for publication. Please find some specific comments as follows.

  1. The abstract should describe novelty of the work more clearly.
  2. There are a number of review papers on CO2 mineralization in literature, such as “A review of carbon dioxide sequestration by mineral carbonation of industrial byproduct gypsum” “CO2 mineral carbonation using industrial solid wastes: A review of recent developments”. This paper should provide more information to justify why this work is different and needed. The comparison of this paper with above newest review papers should be indicated in the Introduction part.
  3. A section containing a techno‐economic analysis with sustainability and social perception is required.
  4. The carbonation of red gypsum (from TiO2 production) is suggested to add in this review.

Author Response

  1. The abstract should describe novelty of the work more clearly.

Response:  

Thanks for the suggestion. In this study, the latest contents of the bench scale study and the point of detailed examining the economic analysis were described. The novelty has been explained as last sentence in abstract.

Line 21–23: “A state-of-the-art bench-scale plant study was also proposed. In addition, economic feasibility was investigated based on a bench-scale study to analyze the future applicability of this technology.”

  1. There are a number of review papers on CO2 mineralization in literature, such as “A review of carbon dioxide sequestration by mineral carbonation of industrial byproduct gypsum” “CO2 mineral carbonation using industrial solid wastes: A review of recent developments”. This paper should provide more information to justify why this work is different and needed. The comparison of this paper with above newest review papers should be indicated in the Introduction part.

Response:

As suggested two references have been cited and discussed in the introduction part.

We focused on direct aqueous carbonation using ammonia and analyzed economic feasibility in detail. Furthermore, there have been few discussions on bench–pilot scale studies of mineral carbonation by reacting flue gas and ammonia with waste gypsum, and there have been no papers that conducted a detailed economic analysis. Comparison with previous studies along with discussion has been added in introduction.

Line 66–78: “Several researchers have recently reported developments in this field. Parameters affecting direct aqueous MC and indirect aqueous MC using industrial byproduct gypsum, including FGD gypsum, phosphogypsum, and red gypsum, have been analyzed by Wang et al. [22]. An analysis of the scale-up applications of MC using industrial solid waste was also reported by Liu et al. [25].

In the present study, the focus was on direct aqueous carbonation using NH3 and the in-detail analysis of its economic feasibility. Recent progress in the carbonation performance of two types of byproduct gypsum, FGD gypsum and phosphogypsum, was evaluated based on CO2 carbonation efficiency. The factors influencing the carbonation kinetics and preparation of CaCO3 were reviewed. Furthermore, there have been limited discussions on bench-scale studies of MC by reacting flue gas and NH3 with waste gypsum, and a detailed economic analysis has not been reported. Economic feasibility was investigated based on a bench-scale study to analyze the future applicability of this technology.”

  1. A section containing a techno‐economic analysis with sustainability and social perception is required.

Response:

Thanks for the advice. We agree that writing about sustainability and social perception of this technology is necessary. Through economic analysis, chapter 5 describes whether this technology is sustainable and in what ways it is promising:

Line 272–279: “The energy to be used for raw material crushing and the CO2 capture process can be saved, as this technology directly reacts with raw materials without pretreatment with flue gas. This is promising because it is carried out by a single process in an aqueous solution [75, 76]. The facility is not structurally complicated; therefore, it is advantageous for scaling up. Depending on the connection with the target plant to be applied, there is the possibility of reducing the steam cost with the operating costs. After the carbonation reaction, a useful material is produced through the crystallization of (NH4)2SO4, thereby improving the economic feasibility of MC technology.”

  1. The carbonation of red gypsum (from TiO2 production) is suggested to add in this review.

Response:

Thank you for your suggestion. It has been reported in numerous papers that red gypsum, a waste product from the titanium dioxide industry, can also sequester CO2.

Waste gypsum (byproduct gypsum) is generated by a different industrial process and it includes the flue gas desulfurization gypsum, phosphogypsum, and titano-gypsum or red gypsum, etc.

The annual production of FGD gypsum and phosphogypsum is more than 140 million tons (Koralegedara et al., 2019; Tan et al., 2018) and 280 (Zhou et al., 2016) million tons, respectively. Meanwhile, the annual output of RG was approximately 18.5 million tons (Liu et al., 2021). If the total amounts of FGD gypsum and phosphogypsum were used to sequester CO2 by MC, almost 100 million tons of CO2 would be stored every year.

Generally, the main component of waste gypsum (byproduct gypsum) is calcium sulfate dihydrate (CaSO4·2H2O) and its purity depends on the production process. The purity of FGDG, PG, and RG is approximately 95% , 90% (Zhao et al., 2015), and 75% (Azdarpour et al., 2014; Rahmani et al., 2016), respectively. 

For these reasons, we have discussed FGD gypsum and phosphogypsum.

Reviewer 2 Report

The manuscript is related to the recycling of industrial waste gypsum using mineral carbonation. Generally, the manuscript may be interesting. However, it is not well written. I would like to expect from the review much more critical point of view. The only one critical analysis in the article is about the mineral and chemical composition of the waste. Therefore I can not recommend this article for publication. I have just also major comments before further processing:

- Introduction – please define abbreviations first time used (e.g. MC, FGD),

- English language should be carefully revised by English Native Speaker. There are many typos and mistakes and thus it is hard to understand the article (e.g. “flue gas” in the abstract),

- in my opinion Fig. 2 is not necessary,

- Review must be deeper. I would like to see more results there as well as the comparison of the results from different sources,

- I suggest to not use such a large citation pockets like [1-7], [12-17] etc. but rather cite each reference individually (e.g. as described in [2]).

Author Response

- Introduction – please define abbreviations first time used (e.g. MC, FGD),

Response:

Thank you for your comments. Following the suggestion, it has been modified to define abbreviation in the first used sentence.

Line 27–28: “Mineral carbonation (MC) plays an important role in carbon dioxide (CO2) storage. Most MC technological developments are still focused on ex situ process concepts.”

Line 41-43: “In the case of flue gas desulfurization (FGD) gypsum, the flue gas of the power plant can be used directly, and desulfurized gypsum, the starting material, can be continuously supplied.”

- English language should be carefully revised by English Native Speaker. There are many typos and mistakes and thus it is hard to understand the article (e.g. “flue gas” in the abstract),

Response:

The entire manuscript was additionally edited through the English editing service.

- in my opinion Fig. 2 is not necessary,

Response:

It is thought that research on a scale larger than the lab scale is necessary for the practical application of this technology. Figure 2 shows the scale of the instrument used in the most recent bench scale study. This study is described in more detail in Chapter 4.

Line 229–240: “Experiments were conducted by manufacturing a bench-scale continuous gypsum carbonation facility, and the reaction system consisted of a bubble column reactor (left of Figure 2) and CO2 and nitrogen (N2) injection facilities (right of Figure 2). The bubble column reactor increased the dissolution efficiency of CO2 such that the flue gas injected into the lower part of the slurry passed through the slurry and moved to the upper part. This study was conducted using FGD gypsum, which has a composition of 32.49 wt% CaO and 44.84 wt% SO3, generated in a thermal power plant, and was used as a raw material for carbonation without pretreatment to improve carbonation reactivity. A slurry with a solid-liquid ratio of 15 wt% containing 120 wt% NH3 based on a chemical equivalent was injected at a rate of 240 kg/h, and 15 vol% of CO2 was injected at a rate of 80 kg/h. Optimization was performed by operating a 1000 ton/y scale facility for 2000 h, resulting in a 97% carbonation rate and crystallization of ((NH4)2SO4).”

Figure 2. Bench-scale continuous gypsum carbonation facility manufactured by KIGAM (left – bubble column reactor; right – CO2 and nitrogen (N2) injection facilities) [70]

- Review must be deeper. I would like to see more results there as well as the comparison of the results from different sources,

Response:

Thanks for the suggestion. In this study, the latest contents of the bench scale study and the point of detailed examining the economic analysis were described.

The results of the bench scale continuous carbonation reactor research were cited in chapter 4, and detailed economic analysis results were further discussed in chapter 5.

- I suggest to not use such a large citation pockets like [1-7], [12-17] etc. but rather cite each reference individually (e.g. as described in [2]).

Response:

Thanks for the suggestion. We are agreeing with the reviewer’s opinion about the modification of the citation. The citation has been modified as reviewer’s recommendation.

Line  28–30: “. Since the 1990s, scientists have begun fundamental research on MC using peridotite to accelerate natural processes [1].”

Line  37–39: “Industrial waste can come from various sources, such as coal fly ash [6-8], metallurgical slag [9-11], waste concrete [12, 13], tailings [13, 14], and industrial waste gypsum [15, 16].”

Reviewer 3 Report

1) In the revised manuscript, the authors significantly modified the presentation of the manuscript along with the title and capex evaluation. After going through the revised manuscript, I would like to suggest for the consideration of this manuscript as a technical note, and essentially not as a review article. There is nothing which allow the manuscript to consider as a review. 

2) Moreover, The authors should provide a supplementary file defining the assumptions and basis of the operational cost is calculated in Table 3.

3) In Fig. 2, the authors should explain the equipment name. How the readers can get any idea about the work by only seeing the photograph without knowing what is that? 

4) This reviewer would like to suggest for a minor revision.

Author Response

1) In the revised manuscript, the authors significantly modified the presentation of the manuscript along with the title and capex evaluation. After going through the revised manuscript, I would like to suggest for the consideration of this manuscript as a technical note, and essentially not as a review article. There is nothing which allow the manuscript to consider as a review.

Response:

Thanks for the recommendation. As recommended, we are willing to change to a technical note.

2) Moreover, The authors should provide a supplementary file defining the assumptions and basis of the operational cost is calculated in Table 3.

Response:

The authors are thankful to the reviewer for the appreciation of the idea. We will submit supplementary material along with the manuscript to explain Table 3.

3) In Fig. 2, the authors should explain the equipment name. How the readers can get any idea about the work by only seeing the photograph without knowing what is that?

Response:

As you advised, we also think we need to improve our explanations. Figure 2 shows the scale of the instrument used, which has no specific name, in the most recent bench scale study. This study is described in more detail in Chapter 4.

Line 229–240: “Experiments were conducted by manufacturing a bench-scale continuous gypsum carbonation facility, and the reaction system consisted of a bubble column reactor (left of Figure 2) and CO2 and nitrogen (N2) injection facilities (right of Figure 2). The bubble column reactor increased the dissolution efficiency of CO2 such that the flue gas injected into the lower part of the slurry passed through the slurry and moved to the upper part. This study was conducted using FGD gypsum, which has a composition of 32.49 wt% CaO and 44.84 wt% SO3, generated in a thermal power plant, and was used as a raw material for carbonation without pretreatment to improve carbonation reactivity. A slurry with a solid-liquid ratio of 15 wt% containing 120 wt% NH3 based on a chemical equivalent was injected at a rate of 240 kg/h, and 15 vol% of CO2 was injected at a rate of 80 kg/h. Optimization was performed by operating a 1000 ton/y scale facility for 2000 h, resulting in a 97% carbonation rate and crystallization of ((NH4)2SO4).”

Figure 2. Bench-scale continuous gypsum carbonation facility manufactured by KIGAM (left – bubble column reactor; right – CO2 and nitrogen (N2) injection facilities) [70]

4) This reviewer would like to suggest for a minor revision.

Response:

The authors thank the reviewers for their appreciation of the ideas. Following the suggestion, the recommendation has been provided in the revised manuscript. The changes have been marked.

Reviewer 4 Report

The article ‘’ Recycling of industrial waste gypsum using mineral carbonation’’ has been presented for the evaluation’’. The article is interested, however, it requires major revisions before it can be considered for the publication in the sustainability journal. In the following some major and minor comments for the authors to rework on the subject matter:

  1. Line 11-12 in the abstract ‘’ Recently, alkaline industrial waste containing a high calcium (Ca) content has been utilized.’’. This line in the abstract looks very general. Abstract should include very specific information related to the problem/ study / solution.
  2. The abstract should include some statistics / figured values of the parameters investigated.
  3. There should be a tabular representation of the block chain type diagram in section 2 ‘’Characterization of waste gypsum’’.
  4. Figure 1 should have reference or citation, if it is taken from the literature. It seems this is not new figure, please check and add accordingly.
  5. What purpose table 1 serves, as it only mention about the Oxide composition for various gases and this is obvious, why this table is necessary. If authors want to add some table from the literature studies related to the oxide weight%, there should be some statistical analysis table about this composition. This gives a reflection of mere literature table in the current form.
  6. Similar is the case with the table 2.
  7. Figure 2 is from the literature. Normally, in the review articles, the authors are supposed to take data from the literature for doing some analysis of the data and to present for a significant value addition. Accordingly, there is no reflection of the same from figure2. Also, what significant value addition is shown from this figure? This is mere a picture of a plant available on internet as well. Authors are advised to take conscious measures regarding representation of visuals / graphs in the research papers.
  8. Line no. 261 says ‘’ the simulated MC processes are shown in Figures 1 and 2, respectively.’’ Whereas figure 2 have no simulation representation. Please clarify and explain accordingly.
  9. Instead of cost evaluation, authors should use the term ‘’techno-economic analysis’’ for better representation of research.
  10. Table 3 just states about the different type of costs involved, whereas the table says ‘’life cycle coasting’’. Authors are advised to do the life cycle assessment for the OPEX and CAPEX of the pilot study. Only mentioning the pilot scale project cost give no outcome. There should be some integrated cost analysis, which can help out the community to get the idea for the commercialization of the process.
  11. There is no conclusion, and without conclusion in fact the article should be rejected. Authors are supposed to follow the structure of a scientific research / review paper. Authors should explain exclusively about the conclusion of this study and its future path / way forward for its better utilization.
  12. Many references are older than 3 years and this is very critical. In the review articles, normally it is desired to get the data of the last 3-4 years and to reflect with the latest developments. Authors are advised to get few references from this journal as well to compare the literature with the relevant of the same journal for the better readership of this journal.

Author Response

The article ‘’ Recycling of industrial waste gypsum using mineral carbonation’’ has been presented for the evaluation’’. The article is interested, however, it requires major revisions before it can be considered for the publication in the sustainability journal. In the following some major and minor comments for the authors to rework on the subject matter:

  1. Line 11-12 in the abstract ‘’ Recently, alkaline industrial waste containing a high calcium (Ca) content has been utilized.’’. This line in the abstract looks very general. Abstract should include very specific information related to the problem/ study / solution.

Response:

Thanks for the suggestion.

The reason why the technology is attracting attention is described (Line 11–14).

  1. The abstract should include some statistics / figured values of the parameters investigated.

Response:

Following the suggestion, it has been modified to suggest some values including CaO content, in each waste gypsum, and conversion ratio (Line 14–16).

  1. There should be a tabular representation of the block chain type diagram in section 2 ‘’Characterization of waste gypsum’’.

Response:

We couldn't provide the analysis you mentioned because it doesn't have the features to form a blockchain. However, the content of Ca and S in the two waste gypsum has different characteristics, so it was introduced.

  1. Figure 1 should have reference or citation, if it is taken from the literature. It seems this is not new figure, please check and add accordingly.

Response:

Figure 1 was prepared by citing gas composition from two literatures. Following the suggestion, it has been modified to suggest references (Line 226–227).

  1. What purpose table 1 serves, as it only mention about the Oxide composition for various gases and this is obvious, why this table is necessary. If authors want to add some table from the literature studies related to the oxide weight%, there should be some statistical analysis table about this composition. This gives a reflection of mere literature table in the current form.

Response:

The table provides constituent for various waste gypsum. Additionally, the content of CaSO4 and CaCO3 was calculated and presented, using the results of each waste gypsum component through XRF analysis (Table 1).

  1. Similar is the case with the table 2.

Response:

The content of CaSO4 and Ca3(PO4)2 was calculated and presented, using the results of each waste gypsum component through XRF analysis (Table 2).

  1. Figure 2 is from the literature. Normally, in the review articles, the authors are supposed to take data from the literature for doing some analysis of the data and to present for a significant value addition. Accordingly, there is no reflection of the same from figure2. Also, what significant value addition is shown from this figure? This is mere a picture of a plant available on internet as well. Authors are advised to take conscious measures regarding representation of visuals / graphs in the research papers.

Response:

It shows the scale of plant study by showing the continuous plant study equipment used in the actual study. Following the suggestion, Equipment names and reaction directions have been indicated in the Figure 2.

  1. Line no. 261 says ‘’ the simulated MC processes are shown in Figures 1 and 2, respectively.’’ Whereas figure 2 have no simulation representation. Please clarify and explain accordingly.

Response:

Following the suggestion, the referenced figure number.

  1. Instead of cost evaluation, authors should use the term ‘’techno-economic analysis’’ for better representation of research.

Response:

Following the suggestion, the term has been modified as your suggestion.

  1. Table 3 just states about the different type of costs involved, whereas the table says ‘’life cycle coasting’’. Authors are advised to do the life cycle assessment for the OPEX and CAPEX of the pilot study. Only mentioning the pilot scale project cost give no outcome. There should be some integrated cost analysis, which can help out the community to get the idea for the commercialization of the process.

Response:

Table 3 is the input details calculated based on the process of removing CO2 by inputting 300,000 tons of CaSO4 per year. CAPEX was calculated by creating piping & instrument diagram of mineral carbonation. The pilot test results were used to calculate the amount of material and utility inputs included in the operating cost. Detailed information is provided in the supplementary material. We introduce the life expectancy and real discount rate of the initial investment facility used in this calculation so that other researchers can refer to them when they want to set different life expectancy and real discount rate, different by country and time of application. Setting a longer life expectancy and a lower real discount rate can significantly increase profitability. And following the suggestion, total CAPEX has been provided with the Table 3 in the revised manuscript.

  1. There is no conclusion, and without conclusion in fact the article should be rejected. Authors are supposed to follow the structure of a scientific research / review paper. Authors should explain exclusively about the conclusion of this study and its future path / way forward for its better utilization.

Response:

Following the suggestion, conclusion has been provided in light of suggestion (Line 301–328).

  1. Many references are older than 3 years and this is very critical. In the review articles, normally it is desired to get the data of the last 3-4 years and to reflect with the latest developments. Authors are advised to get few references from this journal as well to compare the literature with the relevant of the same journal for the better readership of this journal.

Response:

Thanks for the suggestion. We are agreeing with the reviewer’s opinion about the modification of several references. The citation and references have been modified in revised manuscript.

Round 2

Reviewer 1 Report

The format of references should be modified according to the journal's requirements.

Author Response

The format of references should be modified according to the journal's requirements.

Response:

Thanks for the advice.

The format of references has been modified according to the journal's requirements using EndNote (MDPI format).

Reviewer 2 Report

Some remarks have been properly corrected. However, the main remark has not been significantly considered.  Therefore, "I would like to expect from the review much more critical point of view" as stated in previous report. Still, the only one critical analysis in the article is about the mineral and chemical composition of the waste. Therefore I can not support publication of this article.

Author Response

Some remarks have been properly corrected. However, the main remark has not been significantly considered.  Therefore, "I would like to expect from the review much more critical point of view" as stated in previous report. Still, the only one critical analysis in the article is about the mineral and chemical composition of the waste. Therefore I can not support publication of this article.

Response:

Thank you for reviewing our manuscript.

The manuscript has been substantially modified according to the reviewer's opinion.

Reviewer 4 Report

Authors can further improve the article, however, authors have tried to answer the feedback positively and article can be considered for publication after minor improvement.  

Author Response

Thank you for taking the time to review this manuscript.

We have further revised several points suggested by the academic editor.

It is hoped that this manuscript will be more faithful to the journal.

Round 3

Reviewer 2 Report

The changes are not satisfied. There is no still critical review of existing knowledge in the article

Author Response

(The authors gave the same response as above.)
